# Promotion of Lymphangiogenesis by Targeted Delivery of VEGF-C Improves Diabetic Wound Healing

**DOI:** 10.3390/cells12030472

**Published:** 2023-02-01

**Authors:** Lorenz M. Brunner, Yuliang He, Nikola Cousin, Jeannette Scholl, Livia K. Albin, Bianca Schmucki, Sandrin Supersaxo, Gaetana Restivo, Jürg Hafner, Dario Neri, Sabine Werner, Michael Detmar

**Affiliations:** 1Institute of Pharmaceutical Sciences, Swiss Federal Institute of Technology (ETH) Zurich, 8093 Zurich, Switzerland; 2Institute for Dermatology, University Hospital of Zurich, 8091 Zurich, Switzerland; 3Philochem AG, 8112 Otelfingen, Switzerland; 4Department of Biology, Institute of Molecular Health Sciences, Swiss Federal Institute of Technology (ETH) Zurich, 8093 Zurich, Switzerland

**Keywords:** wound healing, diabetes mellitus, inflammation, lymphangiogenesis, VEGF-C

## Abstract

Chronic wounds represent a major therapeutic challenge. Lymphatic vessel function is impaired in chronic ulcers but the role of lymphangiogenesis in wound healing has remained unclear. We found that lymphatic vessels are largely absent from chronic human wounds as evaluated in patient biopsies. Excisional wound healing studies were conducted using transgenic mice with or without an increased number of cutaneous lymphatic vessels, as well as antibody-mediated inhibition of lymphangiogenesis. We found that a lack of lymphatic vessels mediated a proinflammatory wound microenvironment and delayed wound closure, and that the VEGF-C/VEGFR3 signaling axis is required for wound lymphangiogenesis. Treatment of diabetic mice (db/db mice) with the F8–VEGF-C fusion protein that targets the alternatively spliced extra domain A (EDA) of fibronectin, expressed in remodeling tissue, promoted wound healing, and potently induced wound lymphangiogenesis. The treatment also reduced tissue inflammation and exerted beneficial effects on the wound microenvironment, including myofibroblast density and collagen deposition. These findings indicate that activating the lymphatic vasculature might represent a new therapeutic strategy for treating chronic non-healing wounds.

## 1. Introduction

Injury to the skin initiates a highly optimized wound healing process to restore the skin’s function. Wound healing is classified into three dynamic phases: inflammation, new tissue formation, and remodeling [1,2]. However, if a wound does not pass the various stages of healing in an orderly manner and, therefore, does not heal for a long time or at all, it becomes chronic [3]. In healthy skin, the lymphatic vasculature remains quiescent but becomes activated under pathological conditions. It plays a crucial role in the human body; in addition to the drainage of lymph and dietary lipids, it is involved in immune surveillance in tissues [4]. In recent years, immunomodulatory functions of the lymphatic system have been discovered and shown to play key roles in cancer metastasis and chronic inflammatory diseases [5,6,7]. In chronic venous leg ulcers, it has been reported that lymphatic function is impaired [8]. In addition, reduced expression of the vascular endothelial growth factor (VEGF) receptor 3 (R3) has been suggested in chronic ulcers and decubitus wounds in humans, correlating with lower lymphatic density as expected by its specific expression in the lymphatic endothelium [9,10]. Furthermore, lymphatic function is generally affected in type 2 diabetes mellitus due to the interruption of the lymphatic vascular integrity caused by reduced nitric oxide signaling in wounds [11]. Chronic lower limb wounds, commonly referred to as diabetic foot ulcers, are one of the most common complications of diabetes mellitus, with a high recurrence rate representing a significant burden for patients [12,13]. The impaired lymphatic function can lead to increased and sustained inflammation—a hallmark of chronic wounds [14]. Surprisingly, however, it remains unclear to what extent lymphatic vessels (LVs) are involved in wound healing.

The healing of a wound is initiated in the inflammation phase by platelet aggregation and fibrin clot formation. Neutrophils and monocytes, which differentiate later into macrophages, invade the wound during the inflammatory phase. A proper resolution of the inflammatory response is crucial for the complete healing of a wound, and sustained inflammation also aggravates scarring [15]. Subsequently, normal wound healing progresses into the tissue formation phase, where the major cell types, such as fibroblasts and keratinocytes, migrate and proliferate to close the open wound bed. Keratinocytes undergo hyperproliferation and re-epithelialize, whereas fibroblasts proliferate and many of them differentiate into myofibroblasts. An extracellular matrix (ECM) containing large amounts of fibronectin and type III collagen is produced to facilitate cell migration to the open wound. In addition to forming new blood vessels through angiogenesis, granulation tissue is created along with lymphangiogenesis. Transforming growth factor (TGF)-β, fibronectin extra domain A (EDA), platelet-derived growth factor (PDGF), and mechanical stress induce the differentiation of fibroblasts into myofibroblasts [16]. In turn, expression of EDA-fibronectin is induced by TGF-β, and this fibronectin variant interacts with collagens and integrins [17,18,19]. Myofibroblasts, expressing α-smooth muscle actin (α-SMA) and myosin, enable a wound to contract [20,21]. However, in chronic wounds, the migration, proliferation, and differentiation of fibroblasts can be altered [22,23,24]. Moreover, if the fibroblast–myofibroblast differentiation is impaired, diminished ECM production can occur [25]. In the remodeling phase of an acute wound, type III collagen is gradually replaced during the maturation of the granulation tissue by collagen type I, providing increased stiffness and strength to the scar [26,27]. Faster remodeling reduces scar formation, and increased collagen III production through myofibroblasts can lead to fibrosis [28].

Little attention has been paid to wound lymphangiogenesis, the expansion of LVs, that occurs predominantly in inflammation [29]. The binding of VEGF-C to VEGFR3, among others, is the primary driver of post-developmental lymphangiogenesis [30,31]. VEGF-C is considered to be produced by keratinocytes and fibroblasts during homeostasis, and VEGFR3 was reported to be expressed in myofibroblasts [32,33]. Previous studies suggested that promoting lymphangiogenesis through VEGFR3 stimulation inhibits chronic inflammation, and transgenic delivery of VEGF-C reduces acute skin inflammation [34,35]. Furthermore, increased wound lymphangiogenesis alters leukocyte trafficking and matrix remodeling in diabetic wounds [36]. Interestingly, VEGF-C overexpression using an adenoviral vector was able to restore lymphatic flow, induce lymphangiogenesis alongside angiogenesis, and accelerate the healing of diabetic mouse wounds [37,38]. In another study, the decrease in the number of macrophages led to reduced lymphangiogenesis and delayed diabetic wound healing [39]. Macrophages, among many other factors, produce VEGF-C and have widespread functions in wound healing [40,41]. 

Together, these studies point to a great potential for lymphatic activation via VEGF-C in chronic wounds; however, these effects were not separated from angiogenesis. Furthermore, some limitations still have not been overcome, such as the targeted delivery of the protein to the wound site and the generation of a safe and scalable construct applicable for human use. Previously, a humanized VEGF-C fusion antibody with the human monoclonal antibody F8 specifically binding to the epitope of EDA was developed [42,43]. F8–VEGF-C serves as a suitable moiety to target remodeled and inflamed tissue, such as wounds, and has previously been shown to ameliorate chronic colitis and chronic skin inflammation [43,44]. To what extent LVs contribute to remodeling, fibrogenesis, and healing independent of angiogenesis in wounds remains elusive. To thoroughly investigate the role of LVs in wound healing, a full-thickness excisional wound healing model was used to assess lymphangiogenesis in mice. FVB wildtype (WT), K14-sVEGFR3 transgenic (K14-sVEGFR3) mice lacking cutaneous LVs, and K14-VEGF-C transgenic (K14-VEGF-C) mice with an overabundance of LVs in the skin were used [30,45]. The temporal induction of lymphangiogenesis and the importance of the VEGFR3/VEGF-C signaling axis in wounds are presented. Indirect effects on myofibroblast formation and collagen deposition were observed in K14-VEGF-C mice. To confirm these findings and to improve the wound quality in a model relevant for diabetic wounds, targeted VEGF-C delivery was used to induce lymphangiogenesis and activate the LVs. We show that fundamental wound healing mechanisms, such as inflammation, myofibroblast density, and collagen deposition, are affected by activated LVs. Myofibroblasts are recruited at a higher density to wounds, whereas their density was reduced with an expanded lymphatic network later. In summary, this study provides a comprehensive new understanding of the role of LVs in wound healing and suggests that stimulation of lymphangiogenesis is a relevant treatment strategy for chronic wounds in the future.

## 2. Materials and Methods

### 2.1. Animals and Wounding Experiments 

K14-VEGF-C and K14-sVEGFR3 mice, expressing either VEGF-C or sVEGFR3 under control of the keratin 14 promoter (FVB1 genetic background) [30,45], were mated with WT mice, and their F1 progeny was used for wounding experiments and RNA sequencing. WT littermates were used as controls. Diabetic BKS.Cg-Dock7^m^ +/+ Lepr^db^/J (db/db) mice were used (Charles River, Sant’Angelo Lodigiano, Lombardy, Italy). The mice were given at least 14 days of acclimatization and their non-fasting blood glucose levels (>250 mg/dL) were measured before the experiment. The mice were housed under specific opportunistic pathogen-free (SOPF) or opportunistic pathogen-free (OPF) conditions and received food and water ad libitum. The housing was temperature-controlled and had a 12 h/12 h light/dark cycle. 

A full-thickness excisional wound model with female mice was used to study the kinetics of wound healing. In brief, mice (8–10 weeks old) were anesthetized by an intraperitoneal (i.p.) injection of a mixture of ketamine (16 mg/kg) and xylazine (0.8 mg/kg) in phosphate-buffered saline (PBS), the backs were shaved and cleaned with 70% ethanol, and four wounds (5 mm diameter) were generated (two on each side) on the back.

Wound tissues were collected for immunostaining/histology by fixing in 4% phosphate-buffered paraformaldehyde and embedded in paraffin (LogosJ, Milestone Medical, Scano Al Brembo, Bergamo, Italy), or frozen in a tissue freezing medium (O.C.T. Tissue Tek, VWR, Radnor, PA, USA). The wounds were divided along the cranial to the caudal end and embedded so that the sections represented the center of the wound.

### 2.2. Antibody Treatment of Mice

Fusion proteins were produced as described previously [43,44]. Briefly, the F8–VEGF-C stably transfected FreeStyle CHO-S cells (ThermoFisher Scientific, Waltham, MA, USA) were cultured at 37 °C to a concentration of 4 million cells/mL in PowerCHO-2CD medium (Lonza, Basel, Switzerland). The cells were then transferred to Pro-CHO-4 (Lonza) and incubated at 31 °C for 5–7 days. As for the F8-small immunoprotein (SIP), FreeStyle CHO-S cells (ThermoFisher) were cultured at 37 °C to a concentration of 4 million cells/mL in PowerCHO-2CD medium (Lonza) and transiently transfected using PEI (polyethylenimine) in Pro-CHO-4 (Lonza) with plasmids (0.8–0.9 µg/million cells) encoding F8-SIP. At 31 °C, the transfected cells were incubated for 5–7 days. Fusion proteins were purified by protein A affinity chromatography (HiTrap Protein A HP, GE, Healthcare Life Sciences, Chicago, IL, USA) as described previously [46]. The db/db mice received intravenous injections of F8-SIP or F8–VEGF-C (50 µg) into the tail vein every other day from day one post-wounding. The mice were treated with three injections (end of the study on day 7) or five injections (day 10). In additional studies, from the day of wounding, mice received i.p. injections of 800 µg of the rat anti-mouse VEGFR3 blocking antibody (mF4–31C1, ImClone Systems Inc., New York, NY, USA) [47,48] or rat IgG isotype control (Sigma, Munich, Germany) every third day (five injections in total).

### 2.3. Histology and Immunostaining

The standard procedure for hematoxylin and eosin (H&E) staining is explained elsewhere [49]. For Herovici’s stains, deparaffinized slides were immersed in a celestine blue/iron alum solution for 5 min, washed in tap water for 2 min, immersed in a 5% aqueous iron alum/1% alcoholic hematoxylin/4% FeCl_3_/HCl solution for 5 min, washed in tap water for 2 min, followed by a metanil yellow/acetic acid solution for 2 min, immersed in the acetic acid solution for 2 min, washed in tap water for 2 min, dipped in a Li_2_CO_3_ solution for 2 min, immersed in a methyl blue/acid fuchsin/picric acid/Li_2_CO_3_ solution for 2 min, and 1% acetic acid for 2 min [50], and then mounted with Eukitt© (Sigma). For Herovici’s stains, light blue pixels were considered as newly deposited, young collagen and purple pixels as highly crosslinked, mature collagen. 

For immunofluorescence staining of the wounded skin, 7 μm formalin-fixed paraffin-embedded (FFPE) sections were deparaffinized, blocked with a blocking solution (5% donkey serum, 1% bovine serum albumin (BSA), 0.1% Triton-X in PBS) for 1 h, and incubated overnight with goat biotinylated anti-LYVE1 (1:50, BAF 2125, R&D Systems, Minneapolis, MN) and rat anti-CD31 (1:20, DIA-310, Dianova, Hamburg, Germany), goat anti-α-smooth muscle actin (SMA) (1:100, MA511544, Invitrogen, Waltham, MA, USA) and rabbit anti-desmin (1:200, ab13200, Abcam, Cambridge, UK), goat anti-collagen I (1:200, 1310-01, Southern Biotech, Birmingham, AL, USA) and rabbit anti-collagen III (1:100, ab7778, Abcam) antibodies, followed by donkey anti-rat AlexaFluor 488 (1:200, A-21208, ThermoFisher) and donkey anti-rat streptavidin AlexaFluor 594 conjugate (1:200, S32356, Invitrogen), donkey anti-rat AlexaFluor 594 (1:200, A-11058, ThermoFisher) and donkey anti-rabbit AlexaFluor 488 (1:200, A21206, ThermoFisher) or donkey anti-goat AlexaFluor 488 (1:200, A-11055, ThermoFisher) and donkey anti-rabbit AlexaFluor 594 (1:200, A21207, ThermoFisher) secondary antibodies, counterstained with DAPI (Hoechst 33342, Sigma) and mounted with Mowiol 4–88 (475904, Calbiochem, San Diego, CA, USA). Cryosections of 20 μm were fixed in −20 °C acetone for 2 min and 4 °C methanol for 5 min, blocked with a blocking solution for 1 h, and incubated overnight with goat anti-CD45 (1:200, AF114, R&D Systems), rat anti-CD68 (1:100, ab53444, Abcam), and/or rat anti-Ki67 (1:200, 14-5698-80, eBioscience, San Diego, CA, USA), followed by donkey anti-goat AlexaFluor 488 (1:200, A-11055, ThermoFisher), donkey anti-rat AlexaFluor 488, and/or donkey anti-rat AlexaFluor 594 (1:200, A-21209, ThermoFisher) secondary antibodies, counterstained with DAPI and mounted with Mowiol 4-88. EDA-FN staining was performed using a self-produced biotinylated F8 antibody in the SIP format (400 μg/mL) [44] and a primary rat anti-CD31 (1:200, 550274, Becton Dickinson, Franklin Lakes, NJ, USA) antibody followed by streptavidin AlexaFluor 488 (1:200, S32354, Invitrogen) and donkey anti-rat AlexaFluor 594. For immunofluorescence staining of human wounds, 7 μm FFPE or cryosections were processed as above and incubated overnight with goat biotinylated anti-LYVE1 (1:50, BAF 2089, R&D Systems, Minneapolis, MN, USA) and mouse anti-CD31 (1:100, M0823, Dako, Agilent Technologies), followed by donkey anti-rat AlexaFluor 488 (1:200, A-21208, ThermoFisher) and donkey anti-rat streptavidin AlexaFluor 594 conjugate (1:200, S32356, Invitrogen) secondary antibodies, counterstained with DAPI and mounted with Mowiol 4-88.

### 2.4. Image Acquisition and Analysis

The images were acquired with the Pannoramic 250 Slide Scanner (3D Histech, Budapest, Hungary) at 20× magnification. The images were analyzed using Fiji with ImageJ2 version 2.3.0/1.53f (Madison, WI, USA) [51,52]. Wound morphometric analysis was performed using H&E-stained sections to determine the average wound width, the average total length of the basement membrane (BM), and to measure and calculate the hair follicle distance. The total epidermis area from the left to right wound edges was measured and divided by the length of the BM to determine the thickness of the epidermis. Re-epithelialization was calculated in percent by the BM length divided by the total width of the wound. For the quantification of the images of the Herovici-stained tissue sections, the colors of interest were set manually. For the Herovici’s stain, blue pixels were newly deposited, and young collagen and purple pixels were considered highly crosslinked and mature collagen [53]. The same color settings were used for each image. For the Herovici stains, the number of stained pixels was expressed as a percent of the total pixels within the granulation tissue.

The edge of the wound was defined as the interface of hair follicles and the presence of subcutaneous fat tissue, muscle, and granulation tissue. The area 500 µm adjacent to the wound edge was defined as the region of interest, termed the wound edge. LV quantification was performed according to the detailed protocol published previously [49]. In brief, the region of interest was defined as wound edge or granulation tissue. Single-channel images of LYVE1 stainings were thresholded and controlled for CD31^+^LYVE1^+^ vessel-like structures. The empty vessels were filled, and the area fraction was measured. The area of the blood vessels was quantified by subtracting the LYVE1^+^ area from the area of thresholded CD31^+^. Quantification of myofibroblasts was performed accordingly; the desmin^+^ area was removed from the total α-SMA^+^ area [54]. For collagen type I and III, CD45 and CD68 quantification, the region of the granulation tissue was selected, thresholded, and the area fraction was measured. The epidermis was defined and traced by DAPI and the BM to quantify proliferating epidermal cells. Ki67^+^ cells were counted in the designated region and normalized to the length of the BM.

### 2.5. RNA Isolation, Sequencing, and Bioinformatics

RNA was extracted from the total tissue lysates of the wounded and unwounded mouse back skin at specified time points using the miRNeasy mini Kit (Qiagen, Hilden, Germany). The RNA quantity and quality were assessed using a Bioanalyzer (Agilent Technologies, Santa Clara, CA, USA) and subsequently used for generating cDNA libraries. RNA sequencing (single-read 100 bp) using Illumina NovaSeq 6000 was performed by the Functional Genomics Center Zurich (FGCZ) with an average sequencing depth of 30 million reads per sample. Low-quality reads and adaptor sequences were removed using trimmomatic v0.33 [55], after which the remaining reads were aligned with the GRCm38 Mus musculus genome build (Ensembl release 92) with STAR v2.4.2a (Cold Spring Harbor, NY, USA) [56]. The expression matrix was generated using the ‘featureCounts’ function from the Rsubread package v1.26.1 [57], and differential expression analysis was determined with DESeq2 v1.25.5 [58] using a cutoff of fold change ≥1.5 and *p* < 0.05. Gene set enrichment analysis (GSEA v3.0) (Cambridge, MA, USA) was performed to identify enriched biological processes [59].

### 2.6. Human Wounds

Wound biopsies were collected and stored as OCT or FFPE blocks at the department of dermatology of the University Hospital Zurich (USZ) with the assistance of the SKINTEGRITY.CH biobank. All the samples used were surplus materials from routine surgeries.

### 2.7. Statistics

Statistical analysis was performed using PRISM software, version 9 (GraphPad Software Inc., San Diego, CA, USA). The data are shown as mean ± SD or SEM as described in the figure legends. A two-tailed, unpaired Student’s *t*-test (for comparing two groups) or a two-way ANOVA (for repeated measurements) with Sidak’s post-hoc test was performed to determine the statistical significance. Samples that did not pass the Grubbs outlier test were excluded. The differences were considered statistically significant if *p* < 0.05.

## 3. Results

### 3.1. Lymphatic Vessels Are Largely Absent in Human Chronic Ulcers

We first stained for the lymphatic marker LYVE1 and the pan-endothelial marker CD31 to identify (CD31^+^LYVE1^+^ vascular structures) and analyzed the density of LVs in human chronic ulcers and adjacent wound edges.

We observed a significantly reduced number of LVs in chronic ulcer regions compared to the adjacent ulcer edges, respectively (Figure 1A,B). Lymphatic vascular structures appeared to be largely absent in chronic ulcer regions, while the number of LVs was increased next to the edges of the ulcers. Patient information can be found in Appendix A.

### 3.2. Lymphangiogenesis in Wounds of Wildtype, K14-VEGF-C, and K14-sVEGFR3 Mice

We next analyzed the dynamics of lymphangiogenesis in wounds of WT mice and then investigated wound healing in K14-sVEGFR3 and K14-VEGF-C mice. 

We observed a continuous increase in the LV area from the baseline on day 0 to 14 dpw at the wound edge (WE) (0.44 ± 0.19% to 1.09 ± 0.36%) and in the granulation tissue (GT) (0% to 0.66 ± 0.31%) of WT mice (Figure 2A,B). The number of LVs at the WE (4.73 ± 1.89 to 11.06 ± 2.91) followed a similar trend, while we observed a strong increase in the GT (0 to 17.22 ± 3.16). The first LVs in the GT of WT mice were observed at 3 dpw. We detected the strongest increase in the LYVE1+ area in WT mice from day 7 to 10 at the WEs, after the peak of angiogenesis and when most wounds were re-epithelialized. In contrast, a consistent absence of LYVE1+ staining was observed in the wounds of the K14-sVEGFR3 mice, whereas the wound angiogenesis of the K14-sVEGFR3 mice appeared comparable to that of the WT mice (Figure 2A). These results indicate a complete absence of wound lymphangiogenesis in the K14-sVEGFR3 mice. The baseline LV area in the K14-VEGF-C mouse skin (1.94 ± 0.81%) and the LV number (12.75 ± 2.36) were significantly higher than in WT mice. A strong increase in the LYVE1+ area and the number of LVs was observed at the WE and in the GT of the K14-VEGF-C mice. The LV area at the WE of the K14-VEGF-C mice remained stable until 7 dpw. The number of LVs in the GT of the K14-VEGF-C mice also remained comparable to that of the WT wounds, while a substantial increase was observed at the WE at 7 dpw. An increase in the LV area was observed at 14 dpw in the wounds of the K14-VEGF-C mice, with several greatly enlarged LVs observed in all of the K14-VEGF-C mice. Further data for lymphangiogenesis in diabetic mice can be found in the Appendix A. To investigate transcriptional changes, whole wound RNA was isolated from all three genotypes and processed for sequencing. Gene ontology analyses showed an enrichment for genes involved in lymph vessel development in the wounds of the K14-VEGF-C mice, including *Prox1* and VEGFR3 (*flt4*) (Figure 2C), whereas the K14-sVEGFR3 mice had a significantly lower expression of lymphatic development-related genes. Moreover, we found that there were no major differences in the expression levels of matrix metalloproteinase-2 (*Mmp-2*) and *Mmp-9* in the wound tissue between WT mice and K14-sVEGFR3 or K14-VEGF-C mice. Further sequencing data can be found in the Appendix A. These results are in agreement with the morphometric vessel analyses and the important role of VEGF-C and VEGFR3 in wound lymphangiogenesis.

### 3.3. The Lack of Cutaneous Lymphatic Vessels Delays Wound Closure and Blocking VEGFR3 Completely Inhibits Wound Lymphangiogenesis

To assess the effects of the absence of cutaneous LVs in wound healing (Figure 2A,B) in more detail, we next investigated the wounds of the K14-sVEGFR3 mice at 3 and 10 dpw. Hematoxylin and eosin (H&E) stainings were used for morphometric wound analyses (Figure 3A). The K14-sVEGFR3 mice exhibited a significantly reduced wound re-epithelialization at 3 dpw (Figure 3B). In addition, the hair follicle distance was significantly increased in the K14-sVEGFR3 mice with a slightly, but not significantly reduced thickness of the epidermis compared to the WT mice (Figure 3C). The distance of the hair follicles in the K14-sVEGFR3 mice was still significantly increased at day 10, whereas a slightly increased epidermal thickness was observed (Figure 3D). A direct comparison of the genes enriched in gene ontology (GO) analyses showed a strong enrichment in inflammation-related signatures and interferon responses (Figure 3E). 

Since the K14-sVEGFR3 mice did not have any detectable skin LVs before wounding, we next investigated wound healing in WT mice that received a blocking antibody for VEGFR3 every third day from wounding until day 12. Stainings for CD31 and LYVE1 confirmed that there was a significantly reduced LYVE1+ area at the WE, and there were no LVs detected in the GT of the VEGFR3-blocked mice at 14 dpw (Figure 3F–G). These results indicate that the lack of LVs impairs the early closure of wounds and that VEGFR3 is essential for wound lymphangiogenesis.

### 3.4. Reduced Myofibroblast Density and Delayed Collagen Maturation in Wounds of K14-VEGF-C Mice

Since the K14-VEGF-C mice showed an increase in the LV area in wound healing (Figure 2A,B), we next investigated the wound morphometry and quality at 7- and 14-dpw. H&E stainings showed that the hair follicle distance and epidermal thickness were comparable in the WT and K14-VEGF-C wounds (Figure 4A,D,E). Immunostaining for α-SMA and CD31 identifying myofibroblasts (α-SMA^+^CD31^-^ cells) and pericytes (α-SMA^+^CD31^+^ cells) revealed that the K14-VEGF-C mice had a significant decrease in the fraction of myofibroblasts in wounds compared to the WT mice (Figure 4B,F). To assess the potential effects of an increased LV density on the matrix deposition in the wounds, we next analyzed Herovici-stained sections of full-thickness wounds. The K14-VEGF-C mice exhibited a significantly increased fraction of newly deposited (young) collagen (light blue) compared to WT mice, and a slight trend towards a lower fraction of highly crosslinked (mature) collagen (purple) was observed in the K14-VEGF-C mice at 14 dpw (Figure 4C,G). The top 10 significant terms in the GO analysis showed enrichment for VEGF receptor signaling and vessel development (Figure 4H). Together, these results suggest a relationship between LV density, myofibroblast density, and collagen matrix deposition.

### 3.5. The F8–VEGF-C Fusion Protein Is Specific for Regenerating Tissue and Improves Diabetic Wound Healing

Wound-bearing diabetic db/db mice were treated with the F8–VEGF-C fusion protein or a control F8-small immune protein (F8-SIP) every other day from day one until the end of the study at 7 or 10 dpw (Figure 5A,B). Since the F8–VEGF-C fusion protein specifically targets the EDA splice variant of fibronectin in order to achieve wound site-specific delivery of VEGF-C, we first studied the expression in wounds by specific staining for EDA-fibronectin and CD31 in a 5-day db/db wound. Indeed, a strong staining for EDA fibronectin was observed throughout the GT, particularly in perivascular areas (Figure 5C). To measure wound closure in db/db mice, macroscopic images of wounds treated with F8-SIP or F8–VEGF-C were obtained (Figure 5D). Quantification showed a significantly smaller wound area at day three in the F8–VEGF-C-treated mice and a trend towards faster wound closure up to day 10 (Figure 5E). Morphometric analysis of the H&E-stained sections of the full-thickness wounds showed a consistent trend toward improved healing after treatment with the F8–VEGF-C fusion protein (Figure 5F). At day 7, the re-epithelialization was slightly increased, the average wound width slightly reduced, and the epidermal thickness slightly increased in the F8–VEGF-C-treated diabetic wounds compared to the F8-SIP-treated wounds (Figure 5G). The F8–VEGF-C-treated wounds showed a decreased number of proliferating keratinocytes at 7 dpw (Figure 5H), as compared to the F8-SIP-treated wounds, likely reflecting the advanced stage of wound healing.

### 3.6. Targeted Delivery of VEGF-C Potently Induces Lymphangiogenesis in Diabetic Wounds

To investigate the effects of targeted delivery of VEGF-C on lymphangiogenesis in diabetic wounds, the wounds were harvested at 7 and 10 dpw. Immunostaining for CD31 and LYVE1 showed that F8–VEGF-C treatment significantly increased the area and number of LVs in the GT, while the WE area showed similar trends at 7 dpw (Figure 6A,B). The LV size was comparable in both treatment groups (Figure 6C). The blood vessel area (CD31^+^LYVE1^−^) was comparable in both treatment groups (Figure 6D), suggesting that there were no off-target effects of F8–VEGF-C on blood vessel angiogenesis. Furthermore, treatment with either F8-SIP or F8–VEGF-C showed no significant changes in the lymphatic vessel area and number in unwounded skin (Appendix A). The increase in the area and number of LVs in the F8–VEGF-C-treated group was further confirmed in 10-day wounds with an even greater, significant increase, while LVs at the edges of the wound were comparable (Figure 6E). Gene ontology analysis (Figure 6F) revealed that the top 10 enriched terms after F8–VEGF-C treatment include lymph vessel development, endothelium development, and angiogenesis-related terms, in agreement with the observed vascular changes in the treated wounds. Genes related to lymph vessel development were strongly enriched in the F8–VEGF-C-treated group (Figure 6G). Further sequencing results can be found in Appendix A.

### 3.7. Targeted Delivery of VEGF-C Reduces Immune Cell Density in Diabetic Wounds

To investigate whether an increased LV density might be related to reduced leukocyte density in F8–VEGF-C-treated diabetic wounds, immunostaining for CD45 and CD68 was performed (Figure 7A). The CD45+ area was significantly reduced in the F8–VEGF-C-treated wounds (Figure 7B), whereas the CD68+ area was comparable in both groups (Figure 7C), suggesting that increased lymphatic density enhanced the drainage of immune cells from the wounded site, but not specifically inducing macrophage resolution. Expression analysis of the key inflammatory markers, including *Il1a*, *Il1b*, *Il6*, *Tnf*, and *Ifng* showed a reduced expression in the F8–VEGF-C-treated wounds (Figure 7D).

### 3.8. Targeted Delivery of VEGF-C Increases Myofibroblast Density and Collagen I Deposition in Diabetic Wounds

To further study the impact of increased LV density in diabetic wounds on other fundamental wound healing mechanisms, the myofibroblast density (α-SMA + desmin-cells) was assessed by immunostaining for α-SMA and desmin at 7 dpw (Figure 8A). Quantification of the myofibroblast area revealed a significantly increased density in the F8–VEGF-C-treated wounds at day 7, but not at 10 dpw (Figure 8B), suggesting enhanced recruitment with an improved resolution of myofibroblasts with increased lymphatic density in diabetic wounds. GSEA revealed an enriched term of smooth muscle cell migration in the F8–VEGF-C-treated wounds at 7 dpw (Figure 8C), corroborating the results of our immunohistological analyses. Next, immunostaining for collagen types I and III was performed to assess the progression of collagen deposition. The type I collagen density was significantly increased in the F8–VEGF-C-treated wounds at 7 dpw (Figure 8D,F), suggesting accelerated collagen conversion in wounds with increased lymphatic density. The collagen type III density was significantly reduced in wounds treated with F8–VEGF-C (Figure 8E,F).

## 4. Discussion

Several decades ago, it was proposed that lymphatic function is impaired in chronic ulcers [8], and clinical data from chronic human wounds, including decubitus ulcers, suggested that lymphatic vessels are largely absent in chronic wounds [9]. Today’s established primary care for chronic wounds includes compression bandages that aim to improve blood perfusion and release tissue pressure in affected areas [60], and manual lymphatic drainage has also been reported to have a beneficial role in chronic wounds [61]. While it remains to be elucidated whether the lack of lymphatic vessels might be a consequence or one of the causes of impaired wound healing, our findings of a delayed wound closure of acute full-thickness wounds in K14-sVEGFR3 mice, completely lacking cutaneous lymphatic vessels, indicate that lymphatic vessels are necessary for a physiological wound healing process. 

The delayed wound closure in K14-sVEGFR3 mice might have been caused, at least in part, by reduced drainage of inflammatory mediators and cells away from the wound site. Sustained inflammation is a hallmark of chronic wounds [14], and improved resolution of inflammation has been suggested to result in better wound healing [62]. Indeed, in recent years, lymphatic vessel drainage has been recognized as a critical element in inflammation [29], and the promotion of lymphangiogenesis through stimulation of VEGFR3 significantly inhibits chronic skin inflammation [34,35]. Similarly, improved lymphatic clearance and reduced inflammation after VEGF-C stimulation indicated the therapeutic potential of lymphatic activation in chronic inflammatory diseases [63,64,65]. Indeed, our RNA-seq studies of wound tissues obtained from K14-sVEGFR3 mice and their wildtype controls revealed that genes related to the GO term inflammation were upregulated in the wounds of K14-sVEGFR3 mice, reflecting a proinflammatory wound environment in the absence of cutaneous lymphatic vessels and, hence, efficient lymphatic drainage [66]. 

An increasing number of lymphatic-activating molecules have been identified, including VEGF-C and VEGF-D, fibroblast growth factors, platelet-derived growth factors, and others [67]. The absence of lymphatic vessels in the wound granulation tissue of K14-sVEGFR3 mice, together with the complete abrogation of wound lymphangiogenesis in wildtype mice that we treated with a potent VEGFR3 blocking antibody, indicate that activation of VEGFR3, the receptor for VEGF-C and -D, is the main signaling axis required for optimal wound lymphangiogenesis.

While the absence of lymphatic vessels retarded acute wound closure, overexpression of VEGF-C in epidermal keratinocytes of K14-VEGF-C mice did not enhance the closure of acute wounds, despite a potent induction of lymphangiogenesis by the highly active transgene in regenerating epidermis in the granulation tissue and at the wound edges, resulting in an enhanced density and greatly enlarged lymphatic vessels as compared to wildtype mice. These findings are in line with the concept that acute wound healing mechanisms in mice, including the dynamics of lymphatic vessel growth and function, are already optimized and that an overabundance of lymphatic vessels does not provide any further advantage. 

Based on the observed delay of wound closure in K14-sVEGFR3 mice and the reported reduction of lymphatic vessels in chronic diabetic wounds, we next investigated whether the experimental promotion of wound lymphangiogenesis might improve wound healing in a model of delayed diabetic wounds. While one needs to keep in mind that there is currently no animal model available that fully reflects the pathogenesis of chronic diabetic wounds in humans, we used the widely studied obese and hyperglycemic db/db mouse wound healing model because it shows several similarities with chronic diabetic wounds, including delayed closure, diminished cellular proliferation activity, and decreased granulation tissue formation, all of which are clinically relevant in diabetic wound healing [68]. 

Since viral delivery of VEGF-C might be problematic in the clinical setting, whereas applications of VEGF-C protein face challenges in terms of protein stability, high quantity needed, and targeted delivery to the lymphatic vessels, we decided to apply a newly developed therapeutic approach, namely the F8–VEGF-C fusion protein that specifically conveys VEGF-C to sites of inflammation [43]. The F8 antibody targets the alternatively spliced variant extra domain A (EDA) of fibronectin and was originally developed to target solid tumors [42]. EDA fibronectin plays a vital role in inflammation and wound healing [69,70]. During wound healing, EDA fibronectin is strongly upregulated in remodeled tissues but largely absent in the homeostatic skin [17,19,71,72]. In previous biodistribution studies in mice, we found that the F8 antibody coupled with IL-4, as well as the F8–VEGF-C fusion protein, accumulate in inflamed skin and improve chronic skin inflammation [43,73]. 

Our findings that systemic application of the F8–VEGF-C fusion protein in db/db mice resulted in a significantly increased density of lymphatic vessels in the granulation tissue are in agreement with our previous studies of F8–VEGF-C in inflammatory bowel disease, chronic skin inflammation, chronic colitis, and atherosclerosis [43,44,74]. Importantly, F8–VEGF-C treatment also resulted in reduced inflammatory cell density in the wounds, thus providing further evidence for an anti-inflammatory effect of lymphatic vessel activation, which was associated with a trend toward accelerated wound healing.

Chronic wounds are often characterized by excessive degradation of the extracellular matrix (ECM) and impaired collagen deposition [75]. Fibroblasts of the dermis migrate, proliferate, and differentiate into myofibroblasts, producing structural proteins to form a new ECM that replaces the provisional matrix [16]. In chronic wounds, fibroblast differentiation into myofibroblasts is considered impaired [76,77]. Our results show that the myofibroblast density in F8–VEGF-C-treated diabetic mice was significantly increased at 7 dpw, as compared to control-treated mice, but was at a comparable level at 10 dpw. Moreover, we observed an accelerated substitution of collagen type III by collagen type I in diabetic wounds treated with F8–VEGF-C. Faster ECM remodeling has been reported to reduce scar formation, whereas increased collagen type III production by myofibroblasts can lead to fibrosis and even to hypertrophic scars [28,78]. Our findings reveal that targeted delivery of VEGF-C to wounds has effects that go beyond the modulation of lymphatic vessel function and inflammation by modulating distinct wound healing mechanisms to accelerate wound healing processes that influence wound quality and strength. Further studies are needed to elucidate the exact direct or indirect mechanisms by which VEGF-C delivery modulates myofibroblast recruitment and ECM synthesis, and how these effects relate to the quality of the healed wounds. Taken together, activation of lymphatic vessel growth and functions might represent a promising new approach for the management of chronic non-healing wounds.

## 5. Patents

The authors declare the following competing financial interest(s): D.N. is a shareholder and board member of Philogen, a biotech company that shares commercialization rights for F8–VEGF-C with ETH Zurich. M.D. and D.N. are inventors of the patent US2021163579 (A1) for F8–VEGF-C. The other authors declare no competing financial interest.

## Figures and Tables

**Figure 1 cells-12-00472-f001:**
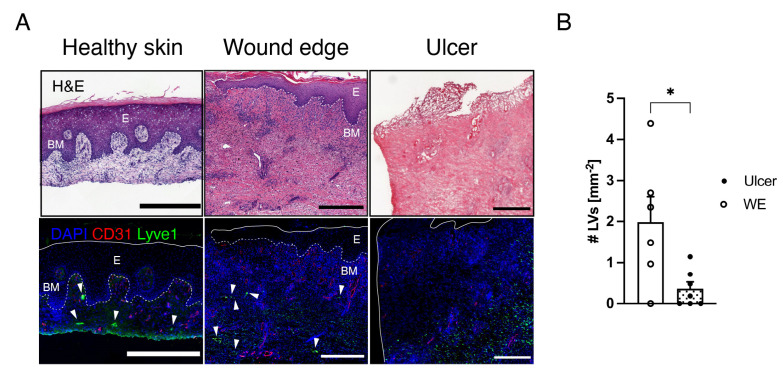
Lymphatic vessels are largely absent in human chronic ulcers. (**A**) Representative H&E and LYVE1 and CD31 stained sections of human healthy skin, wound edge, and ulcer of a chronic wound. Healthy skin is characterized by a continuous basal membrane. The adjacent wound edge is re-epithelialized. The ulcer is defined through its non-epithelialized characteristics. Scale bars: 500 µm. White arrowheads depict LVs. (**B**) Quantification of the number of LVs per mm^2^ in representative regions of wound edges adjacent to ulcerative regions and ulcers. N = 6–7. Students *t*-test, *p* < 0.05 (*).

**Figure 2 cells-12-00472-f002:**
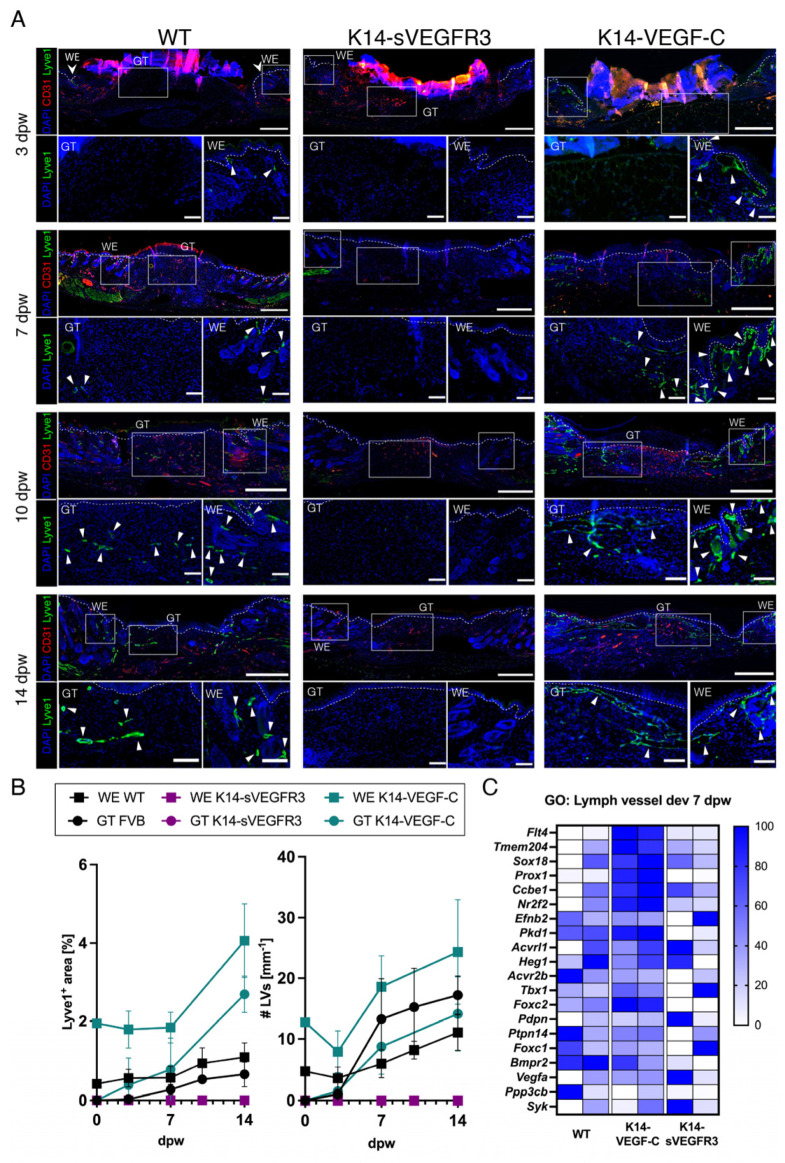
Wound lymphangiogenesis is mediated by the VEGF-C/VEGFR3 signaling axis. (**A**) Representative LYVE1 and CD31 stained sections of unwounded skin, and wounds at 3-, 7-, 10-, and 14-days post-wounding (dpw) of WT, K14-VEGFR3-Fc, and K14-VEGF-C mice. Scale bars: Overview 500 µm, GT, WE: 100 µm. White arrowheads depict LVs. (**B**) LYVE1+ area [%] and the number of LVs divided by the length of BM [mm^−1^] of WT, K14-VEGFR3-Fc, and K14-VEGF-C mice at 3-, 7-, 10-, and 14-days dpw. N = 3–5 mice per group, 2–3 wounds each. (**C**) Relative gene expression of genes in GO: Lymph vessel development. Gene expression is displayed as a percentage of the expression level of the sample with the highest value for each gene. N = 2 wounds per group. Scale: 0–100%. GT: Granulation tissue, WE: Wound edge, dpw: days post-wounding.

**Figure 3 cells-12-00472-f003:**
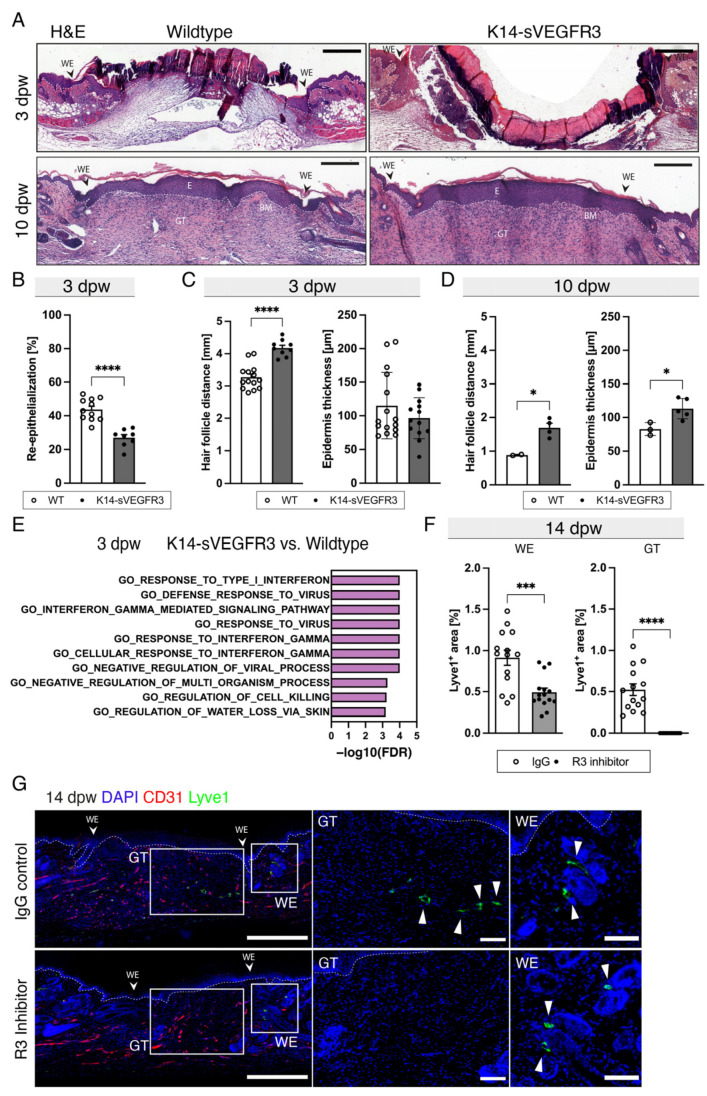
Lack of lymphangiogenesis delays wound closure and blocking of VEGFR3 completely inhibits lymphangiogenesis. (**A**) H&E representative sections of WT and K14-VEGFR3-Fc transgenic mice at 3- and 10-dpw. Scale bar: 500 µm. 8–11 wounds of 5 mice. (**B**) Re-epithelialization [%] of WT and K14-VEGFR3-Fc transgenic mice at 3 dpw. (**C**) Hair follicle distance [mm] and epidermis thickness [µm] of WT and K14-VEGFR3-Fc transgenic mice at 3 dpw. Scale bars: 500 µm. N = 13–16 wounds of 5 mice. (**D**) Hair follicle distance [mm] and epidermis thickness [µm] of WT and K14-VEGFR3-Fc transgenic mice 10 dpw. N = 2–5 mice with 2–3 wounds each. (**E**) Top 10 enriched GO terms in K14-VEGFR3-Fc transgenic versus WT mice at 3 dpw. (**F**) LYVE1+ area [%] and the number of LVs [mm^−1^] 14 dpw. N = 14 wounds of 6 mice. (**G**) Representative images of LYVE1-CD31 stained wound sections 14 dpw. Scale bars: Overview 500 µm, GT, WE: 200 µm. White arrowheads depict LVs. GT: Granulation tissue, WE: Wound edge, dpw: days post-wounding, BM: Basal membrane, E: Epidermis, HF: Hair follicle. Students *t*-test, *p* < 0.05 (*), *p* < 0.001 (***), *p* < 0.0001 (****).

**Figure 4 cells-12-00472-f004:**
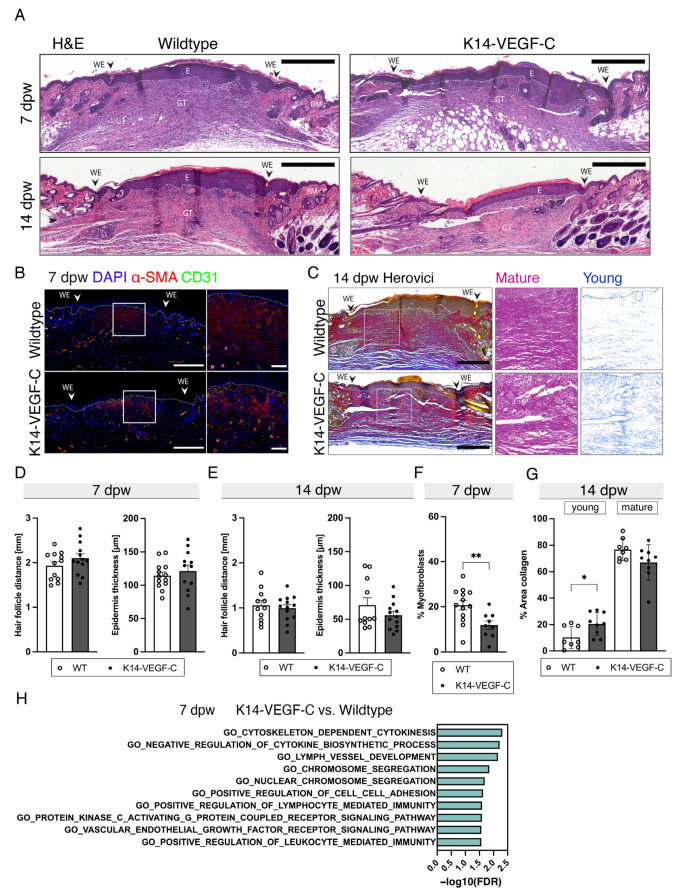
Increased lymphatic vessel area reduces myofibroblast density and delays collagen maturation. (**A**) H&E representative sections of WT and K14-VEGFR3-Fc transgenic mice at 3- and 10 dpw. Scale bar: 500 µm. (**B**) Representative α-SMA and CD31 stained sections of WT and K14-VEGF-C mice 7 dpw. Scale bars: Overview: 500 µm, Zoom-in: 100 µm. (**C**) Herovici’s stain representative sections of WT and K14-VEGF-C mice 14 dpw. Single-channel images of Herovici staining for the colors of mature (purple) and young (light blue) collagen are shown. Scale bars: Overview: 500 µm, mature, young: 100 µm. (**D**) Hair follicle distance [mm] and epidermis thickness [µm] of WT and K14-VEGF-C mice at 7 dpw. N = 12–13 wounds of 5 mice. Scale bars: 500 µm. (**E**) Hair follicle distance [mm] and epidermis thickness [µm] of WT and K14-VEGF-C mice at 14 dpw. N = 11–13 wounds of 5 mice. (**F**) Myofibroblast+ area [%] (CD31+ area deducted from α-SMA+ area) of WT and K14-VEGF-C mice 7 dpw. N = 9–13 wounds of 5 mice. (**G**) Young (light blue) and mature (purple) collagen area normalized to BM of WT and K14-VEGF-C mice 14 dpw. 8–9 wounds of 3 mice. (**H**) Top 10 enriched GO terms in K14-VEGF-C versus WT mice at 7 dpw. N = 2 wounds per group. GT: Granulation tissue, WE: Wound edge, dpw: days post-wounding, BM: Basal membrane, E: Epidermis, HF: Hair follicle. Students *t*-test, *p* < 0.05 (*), *p* < 0.01 (**).

**Figure 5 cells-12-00472-f005:**
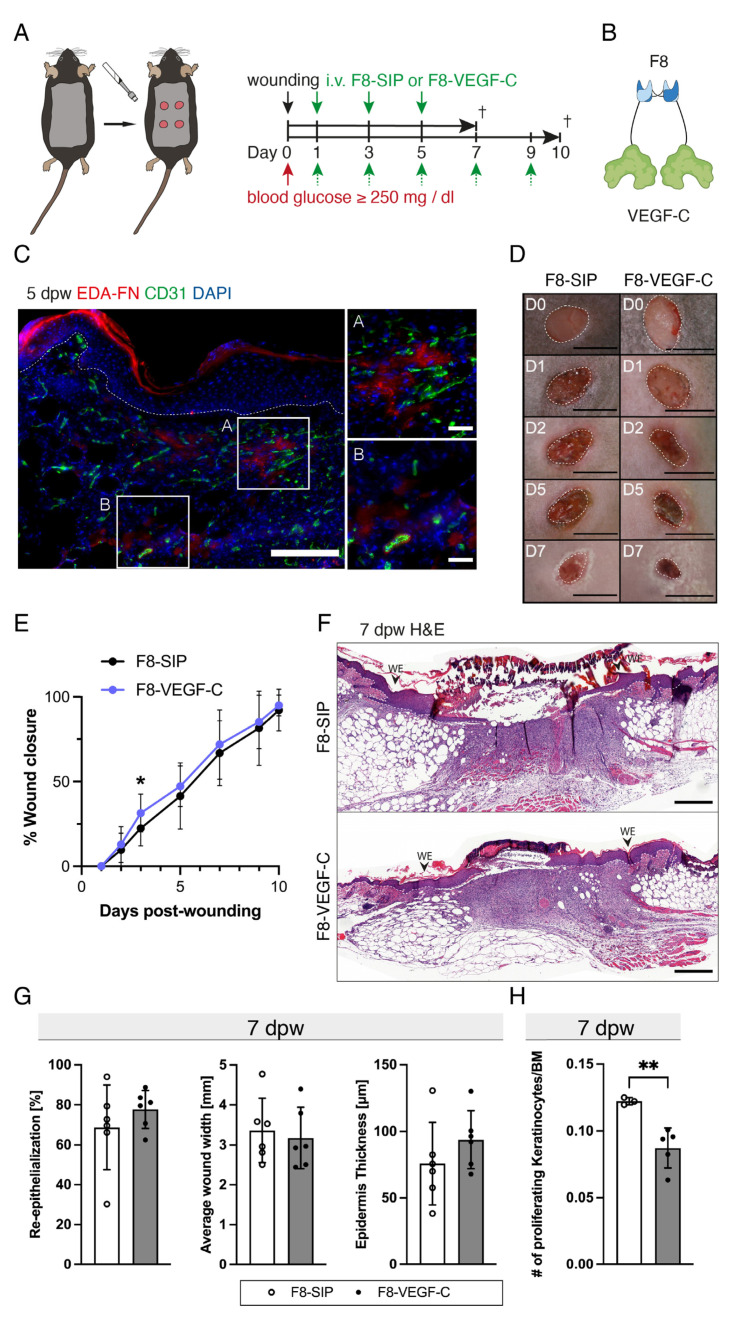
The F8–VEGF-C fusion protein is specific for regenerating tissue and improves wound morphometry. (**A**) Full-thickness excisional wound healing model in db/db mice. Blood glucose measurement prior to wounding on day 0 (inclusion criteria: blood glucose ≥250 mg/dL). Injection of F8-SIP or F8–VEGF-C every other day from day 1. (**B**) Schematic illustration of the fusion protein F8–VEGF-C. (**C**) Representative EDA-FN and CD31 staining image at the wound edge at 5 dpw. Scale bar: Overview 200 µm, Zoom-ins: 40 µm. (**D**) Representative images of wounds of F8-SIP or F8–VEGF-C treated mice at 0, 1, 2, 5 and 7 dpw. Scale bars: 5 mm. (**E**) Wound closure [%] of F8-SIP vs. F8–VEGF-C-treated mice. Wound closure was significantly increased at 3 dpw. N = 15 mice, from 3 independent experiments. (**F**) Representative images of H&E-stained sections of wounds treated with F8-SIP or F8–VEGF-C at 7 dpw. (**G**) Dynamic wound analysis of H&E-stained sections 7 dpw. Slightly increased re-epithelialization [%] and average wound width [mm], and slightly reduced epidermis thickness [µm] in F8–VEGF-C-treated mice. N = 16–18 wounds of 6 mice per group. (**H**) The number of proliferating keratinocytes per µm BM was significantly increased in F8-SIP-treated wounds at 7 dpw. N = 1–3 wound of 5 mice for F8–VEGF-C, N = 2–3 wounds of 3 mice for F8-SIP. BM: basal membrane, dashed lines: basal membrane, WE: Wound edge, dpw: days post-wounding. Students *t*-test, *p* < 0.05 (*), *p* < 0.01 (**).

**Figure 6 cells-12-00472-f006:**
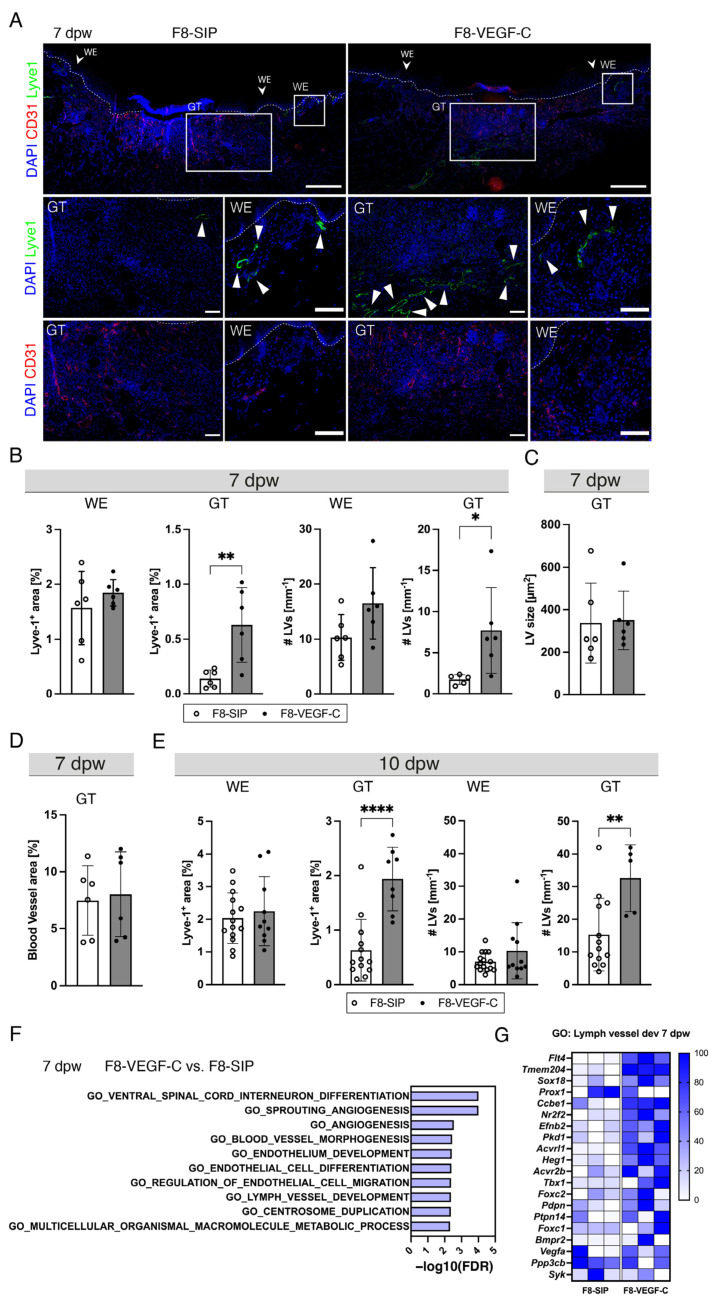
Targeted delivery of VEGF-C potently induces lymphangiogenesis in diabetic wounds. (**A**) Representative images of LYVE1 and CD31 stained sections of F8-SIP and F8–VEGF-C-treated db/db mice at 7 dpw. Magnifications of GT and WE are shown in LYVE1 and CD31, respectively. More LVs are present in the GT area of the F8–VEGF-C treated mice. The CD31+ area was comparable in the GT and the WEs in both groups. White arrowheads depict LVs. The white box indicates areas for magnified images. Scale bars: Overview 500 µm, GT, WE: 200 µm. (**B**) LYVE1+ area [%] and the number of LVs [mm^−1^] at 7 dpw quantified next to the WE and in the GT. Quantification revealed a significant increase of LV area in the GT. The number of LVs per mm BM was significantly increased in the GT of F8–VEGF-C-treated wounds. N = 6 mice per group. (**C**) The LV size was comparable in the two treatment groups in the GT and in the WEs. N = 6 mice per group. (**D**) There was no difference in the blood vessel area between the two treatment groups for the GT area and the WEs at 7 dpw. N = 6 mice per group. (**E**) LYVE1+ area [%] and the number of LVs [mm^−1^] at 10 dpw quantified next to the WE and in the GT. A significant increase in the LV area and number was observed in the GT at dpw. N = 8 mice per group. (**F**) Top 10 enriched GO terms in F8–VEGF-C-treated versus F8-SIP-treated db/db mice at 7 dpw. N = 3 wounds per group. (**G**) Relative gene expression of genes in GO: Lymph vessel development. Gene expression is displayed as a percentage of the expression level of the sample with the highest value for each gene. N = 3 wounds per group. Scale: 0–100%. GT: Granulation tissue, WE: Wound edge, dpw: days post-wounding, BM: Basal membrane, E: Epidermis, HF: Hair follicle. Students *t*-test, *p* < 0.05 (*), *p* < 0.01 (**), *p* < 0.0001 (****).

**Figure 7 cells-12-00472-f007:**
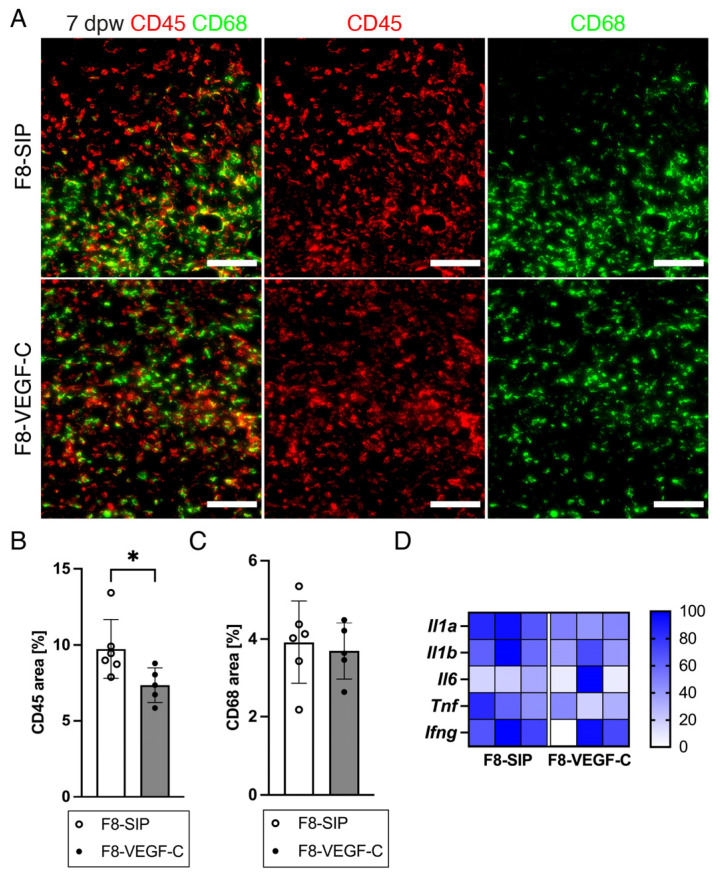
Targeted delivery of VEGF-C reduces immune cell density in diabetic wounds. (**A**) Representative images of CD45 and CD68 stained sections of F8-SIP- and F8–VEGF-C-treated db/db mice at 7 dpw. Scale bars: 200 µm. (**B**) CD45+ area [%] quantified in the GT at 7 dpw. The CD45 area was significantly decreased in the wounds of F8–VEGF-C-treated mice. (**C**) CD68+ area [%] quantified in the GT at 7 dpw. The CD68 area was comparable between the two treatment groups. (**D**) Relative gene expression of selected inflammatory genes from GSEA. Gene expression is displayed as a percentage of the expression level of the sample with the highest value for each gene. N = 9–11 wounds of 5–6 mice per group. Scale: 0–100%. Students *t*-test, *p* < 0.05 (*).

**Figure 8 cells-12-00472-f008:**
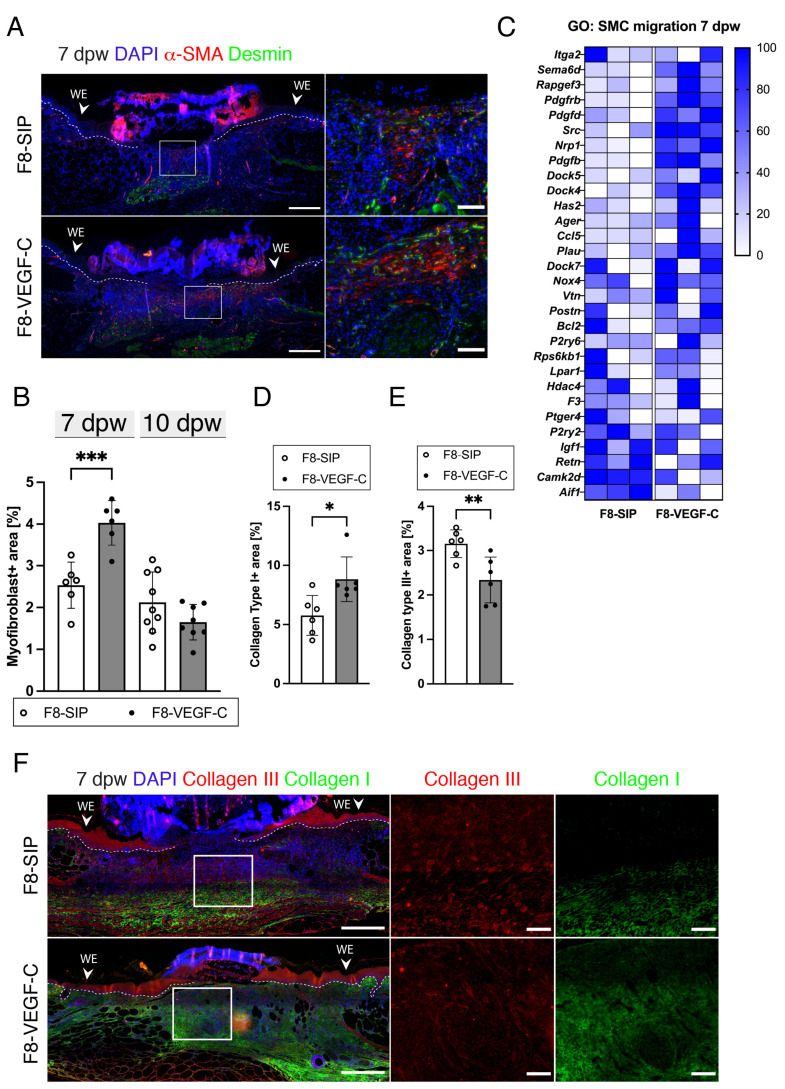
Targeted delivery of VEGF-C increases myofibroblast density and collagen I deposition in diabetic wounds. (**A**) Representative α-SMA and desmin stained sections of F8-SIP- and F8–VEGF-C-treated db/db mice at 7 dpw. Enlarged images showing more abundant myofibroblasts in the center of the F8–VEGF-C-treated wounds compared to the F8-SIP-treated wounds. Scale bars: Overview: 500 µm, Zoom-in: 200 µm. (**B**) Myofibroblast+ area [%] (desmin+ area deducted from α-SMA+ area) of F8-SIP- and F8–VEGF-C-treated db/db mice at 7 dpw. The density of myofibroblasts in the GT of the F8–VEGF-C group was significantly higher compared to the F8-SIP-treated wounds at 7 dpw. The myofibroblast density was comparable at 10 dpw. (**C**) Relative gene expression of genes in GO: SMC migration. Gene expression is displayed as a percentage of the expression level of the sample with the highest value for each gene. N = 3 wounds per group. (**D**) Collagen I+ area [%] at 7 dpw quantified in the GT. The area of collagen type I was significantly increased in the F8–VEGF-C-treated wounds. (**E**) Collagen III+ area [%] at 7 dpw quantified in the GT. The collagen type III area was significantly decreased in the F8–VEGF-C-treated wounds. (**F**) Representative collagen type I and III-stained sections of F8-SIP- and F8–VEGF-C-treated wounds at 7 dpw. The enlarged images highlight that there was more collagen type III and less collagen Type I present in the F8-SIP-treated wounds at 7 dpw. (**B**,**D**–**F**). N = 18 wounds of 6 mice for each group. WE: Wound edge, dpw: days post-wounding, BM: Basal membrane (dashed white line). Students *t*-test, *p* < 0.05 (*), *p* < 0.01 (**), *p* < 0.001 (***).

## Data Availability

Bulk RNA-seq raw data are available at ArrayExpress under accession number E-MTAB-12545.

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
