# Peer review of "Promotion of Lymphangiogenesis by Targeted Delivery of VEGF-C Improves Diabetic Wound Healing"

_cells, 2023, doi:10.3390/cells12030472_

Round 1

Reviewer 1 Report

This research manuscript tried to address the valid question in the field of chronic ulcers healing in diabetes mellitus subjects. Authors have aptly designed the study and conducted the experiments with their previously established protocols and mouse model system and finally presented in a very well written manuscript. However, the outcome interpretations presented by authors are little not clear in every results section. I have some minor suggestions/comments that needs to be addressed, are below

·         Suggest to include Inflammation and Diabetes mellitus as Keywords, so that there will be wider visibility of the study

·         Under materials and methods section, lines 118 to 132 are irrelevant to this study.

·         In the methodology section, under “animals and wounding experiments” method, is there any reason for choosing only female mice for the experiments, if so explain it or substantiate it.

·         In line 142-143, how can it be abbreviated “opportunistic pathogen-free” as “OHB”, any explanation or typo?

·         Lines 161-163 are repeated again in lines 168-169 under 2.2 antibody treatment of mice, rectify it.

·         Throughout the manuscript figures, the “white arrow heads” are sparingly used, however, it has not been defined anywhere else.

·         If possible, I would suggest to check the BM degradation markers such as MMP2 and MMP9 that would strengthen the degradation of BM and recover BM after treatment, as in the 3.1 and 3.2 results sections.

·         Under section 3.2, line 292-293, abbreviating the “mice that lack cutaneous lymphatic vessels (LVs, K14-sVEGFR3 transgenic mice) and in mice that have an increased density of LVs in their skin (K14-VEGF-C transgenic mice)”, is not uniform. Rectify it.

·         In section 3.3, the statistical significance of the epidermal thickness between wild type and sVEGFR3 was not mentioned. If significance is not appropriate, mention not significant (NS).

·         In section 3.6, the off-target effect of F8-VEGF-C is not clearly explained with supportive statements of targeted delivery.

·         The methodology employed for performing the expression analysis of various inflammatory cytokines was not clearly explained in the 3.7 section. Further, an estimation of inflammatory cytokine levels in the animal serum after F8-VEGF-C treatment would be more appropriate.

·         In lines 313-314, substantiate the observation of unusual expansion of LVs in VEGF-C wounds at 14 dpw. If there is no justification for this observation, rather it would be better to remove it from the main figure and can be given as supplementary information.

·         Check for the English usage in line 365, “quality at 7- and 14 dpw”.

·         In figure 4C, the mature and young images are not been properly represented. Is it H&E or IHC, if so, explain it?

·         In lines 481-483 and 580-581, "Increased density of F8-VEGF-C-treated wounds at day 7 but not at 10 dpw" these results needs proper justification because if it is not so significant at 10 dpw, how do you justify it has some therapeutic implications.

Overall, I would appreciate the idea that the systemic application of the F8-VEGF-C fusion protein in db/db mice model resulted in a significantly increased density of lymphatic vessels which in turn helps in wound healing of chronic ulcers.

Reviewer 2 Report

The study examines the role of lymphangiogenesis, the formation of new lymphatic vessels, in wound healing. The study found that stimulation of lymphangiogenesis via the VEGF-C and VEGFR3 signaling axis is necessary for optimal wound healing, and that a lack of lymphatic vessels in the wound granulation tissue can delay wound closure. The study also examines the potential therapeutic benefits of promoting lymphangiogenesis in diabetic wounds, finding that targeted delivery of VEGF-C can improve wound healing and reduce inflammation in a mouse model of diabetic wounds.  The data and conclusions are clearly presented, and the discussion effectively ties together the findings and their implications. Overall, this study makes a valuable contribution to the field and is worthy of publication.

Round 2

Reviewer 1 Report

I agree and accept all the responses submitted by the authors.